# Infrared Image-Enhancement Algorithm for Weak Targets in Complex Backgrounds

**DOI:** 10.3390/s23136215

**Published:** 2023-07-07

**Authors:** Yingchao Li, Lianji Ma, Shuai Yang, Qiang Fu, Hongyu Sun, Chao Wang

**Affiliations:** National and Local Joint Engineering Research Center of Space Optoelectronics Technology, Changchun University of Science and Technology, Changchun 130022, China; liyingchao@cust.edu.cn (Y.L.); fuqiang@cust.edu.cn (Q.F.); ccshy@mails.cust.edu.cn (H.S.); wangchao2017@cust.edu.cn (C.W.)

**Keywords:** background suppression, detection, filtering, infrared, small target enhancements

## Abstract

Infrared small-target enhancement in complex contexts is one of the key technologies for infrared search and tracking systems. The effect of enhancement directly determines the reliability of the monitoring equipment. To address the problem of the low signal-to-noise ratio of small infrared moving targets in complex backgrounds and the poor effect of traditional enhancement algorithms, an accurate enhancement method for small infrared moving targets based on two-channel information is proposed. For a single frame, a modified curvature filter is used in the A channel to weaken the background while an improved PM model is used to enhance the target, and a modified band-pass filter is used in the B channel for coarse enhancement followed by a local contrast algorithm for fine enhancement, based on which a weighted superposition algorithm is used to extract a single-frame candidate target. The results of the experimental data analysis prove that the method has a good enhancement effect and robustness for small IR motion target enhancement in complex backgrounds, and it outperforms other advanced algorithms by about 43.7% in ROC.

## 1. Introduction

Infrared small-target detection is one of the key technologies for infrared detection systems, infrared early warning systems, precision-guidance systems, satellite remote sensing systems, etc. [1,2] The performance of the whole system depends to a large extent on the accuracy of the target enhancement results. Infrared small targets have many inherent properties due to their special imaging mechanisms, which also poses great difficulties for the detection technology. For example, the target size is usually small (ranging from 2 × 2 to over 10 × 10 pixels) due to the long imaging distance; small targets have little texture information and are usually irregular in shape; and the contrast between target and background varies greatly due to the type of target and imaging distance [3,4,5]. In addition, when there is a lot of noise and complex clutter, the real target may be drowned out in the background, which creates a major detection technical challenge. The accurate detection of small targets in infrared motion is therefore a challenge, especially in complex background conditions [6,7]. Complex backgrounds include man-made buildings and natural environments, where the natural environment includes surface vegetation, mountains, etc., which have a rich textural structure with high IR (Infrared Radiation) reflectivity but inherently low IR. Other man-made structures, such as pavements, signs, and buildings, vary greatly in shape and infrared properties. The simultaneous presence of these factors in an IR image can cause significant interference with small-target detection.

Traditional infrared small-target enhancement algorithms mainly include Top-Hat [8,9], Max-Mean, and Max-Median filtering algorithms [10,11], wavelet transform algorithms [12,13], etc. These algorithms have good enhancement effects for infrared images with simple backgrounds but a high false-alarm rate for infrared images with complex backgrounds. An infrared small-target detection algorithm includes single-frame detection and multi-frame detection. Single-frame detection includes a filtering-based detection algorithm, a human-visual-system-based detection algorithm and deep-learning-based detection algorithm. Among many filtering-based detection algorithms, Zhang [14] improved the Robinson Guard filter, which has a high computational efficiency, but the filtering results are easily affected by small targets or interference clutter, and the robustness is poor. Based on the analysis of edge direction, Bae [15] proposed a small infrared target-detection method based on a bilateral filter (BF). This algorithm applied a BF filter as a background predictor for small-target detection. At the same time, compared with the traditional algorithm, it also improves the robustness and detection efficiency, but the high-frequency noise in the infrared image cannot be cleanly filtered out, and only the low-frequency information can be better filtered. Xie [16] proposed a new method based on bidimensional empirical mode decomposition (BEMD), which can effectively suppress background noise and improve the signal-to-noise ratio of original images. However, due to the increase in algorithm complexity, real-time performance is seriously insufficient among many detection algorithms based on the human visual system. Cai [17] proposed an infrared small-target detection algorithm based on the visual contrast mechanism, which uses the local contrast measure (LCM) operator to obtain the significant region, but the algorithm has limited performance on noise and background suppression. In the detection algorithm based on deep learning, LIN [18] designed a 7-layer deep convolutional neural network (CNN) to realize the automatic feature extraction of infrared small targets and the end-to-end suppression of background clutter. However, the biggest limitation of applying deep learning to infrared small-target detection is that the size of the infrared small target to be detected is too small. This makes it difficult for the detector to extract effective features, resulting in the poor detection of small targets and high cost. In this paper, a new method is proposed to enhance small infrared moving targets in complex backgrounds based on two-channel information. Different models of IR small targets in complex backgrounds lead to different causes of false alarms, and the IR small-target images can be processed according to different models to remove false alarms.

The rest of the paper is organized as follows: In Section 2, the steps of the method are described. The theory and methodology of our work will be described in detail in Section 3. The performance of our method and other methods will be analyzed in Section 4. Finally, in Section 5, the work of this paper is summarized.

## 2. Framework

Our approach consists of four main steps, the flow chart of which is shown in Figure 1.

Step 1: Using the developed structural elements, the initial image is closed to remove salient noise and retain as much of the target as possible.

Step 2: The improved curvature filtering algorithm and the PM (Perona-Malik) model algorithm are used in turn to obtain targets that contain candidates based on layer information.

Step 3: The targets containing the candidates based on the space and frequency domain information are extracted by the improved bandpass filtering algorithm and the improved local contrast algorithm in turn.

Step 4: A weighted overlay is used to combine the enhanced images from steps 2 and 3, resulting in the enhancement of a weak target against a complex background in a single frame.

## 3. Methods

In this section, we will discuss the proposed method in detail. The whole enhancement process can be divided into four main parts: pre-processing, A-channel calculation, B-channel calculation, and dual-channel fusion.

### 3.1. Image Pre-Processing

The closure operation is a method of morphological filtering that involves the expansion of the image before the erosion of the image [19,20,21]. The advantage of the closure operation is that it can fill in small cracks and concave corners in the image, enhancing the connectivity and integrity of the image. However, unreasonable choices of the size and shape of structural elements can alter the position and shape of the image, resulting in blurred edges or a loss of detail.

To address the above issues, this paper designed a new structure element based on the shape of the small target. A circular-like template was chosen (in order to ensure that the shape of the graph does not change substantially, the size of the template should not be too large; we usually choose a template of size 5 × 5 or 7 × 7), and the small target structure and small target-like structure elements were divided, as shown in Figure 2:

The closing operations are as follows:(1)U·S=(U⊕S)⊖S
where U is the original image, *S* is the convolution kernel, ⊕ is the expansion operation, and ⊖ is the corrosion operation
(2)$$\mathrm{dst}(x,y)=\max\underset{\left(x^{\prime },y^{\prime }\right);\;=\;\mathrm{element}\;\left(x^{\prime },y^{\prime }\right)\neq 0}{\left[\mathrm{src}\left(x+x^{\prime },y+y^{\prime }\right)\right]}$$
(3)$$\mathrm{dst}\prime (x,y)=\min\underset{\left(x^{\prime },y^{\prime }\right);\;=\;\mathrm{element}\;\left(x^{\prime },y^{\prime }\right)\neq 0}{\left[\mathrm{src}\left(x+x^{\prime },y+y^{\prime }\right)\right]}$$
where dst(x,y) is the image after the expansion operation, dst′(x,y) is the image after the erosion operation, max is the maximum value operation, min is the minimum value operation, and $\mathrm{src}\left(x,y\right)$ is the original image.

### 3.2. A-Channel Layer-Based Overall Enhancement

A single infrared image can be regarded as a target layer, a background layer, and a noise layer, which were filtered according to their layer characteristics, and then the type A target to be verified was accurately extracted to obtain a candidate target in the spatial domain [22]. For the background layer, we used a modified curvature filter for background suppression. This method can effectively suppress the background while removing the noise. Figure 3 shows two classical IR images of small targets with complex backgrounds.

#### 3.2.1. Background-Layer Suppression

The suppression of the background layer is essential to suppressing the target, and in this subsection, we used an improved PM diffusion model algorithm to enhance the target.

Traditional isotropic filtering tends to remove noise while blurring the edges of the image and losing image details. To overcome this drawback, the PM diffusion model was proposed in 1990 [23]. The PM diffusion model is a non-linear diffusion method with the expression:(4)$$\left\{\begin{array}{l} \frac{\partial U(x,y,t)}{\partial t}=\mathrm{div}[g(\nabla U)\cdot \nabla U]\\ U(t=0)=U_{0} \end{array}\right.$$

In Equation (4), *U* is the grey-scale image; div is the scatter operator; ∇ is the gradient operator; and g(∇U) is the edge stop function coefficient, which adaptively derives its smoothing coefficient according to the gradient relationship in different directions around the image pixel points, taking differential smoothing according to the characteristics of the image.

The classical edge stop function [23] is denoted as
(5)$$\left\{\begin{array}{l} g_{1}(\nabla U)=\frac{1}{1+{\left(|\nabla U|/k\right)}^{2}}\\ g_{2}(\nabla U)=\exp\left[-{\left(|\nabla U|/k\right)}^{2}\right] \end{array}\right.$$
where *k* is the gradient coefficient and is a constant greater than 0.

In an IR image, the abrupt regions with large gradients are the target, while the smooth regions with small gradients are the background. Since the PM diffusion model is used to remove noise from the image to make it smooth, and since the IR weak target enhancement requires the suppression of smooth and non-smooth background areas and the retention of abrupt target areas, the diffusion coefficients of the PM model need to be modified as follows: the diffusion coefficients are suppressed at backgrounds with small gradients, while the diffusion coefficients are enhanced at targets with large gradients. In this paper, the diffusion coefficients of suspicious areas where targets may be present were modified by combining the gray scale values of the background pixels to achieve better background suppression while enhancing the target energy information.

The PM model based on small target information enhancement is denoted as
(6)$$\left\{\begin{array}{l} \frac{\partial U(x,y,t)}{\partial t}=\mathrm{div}[g_{3}(\nabla U)\cdot \nabla U]+\frac{U}{\lambda }\\ U(t=0)=U_{0} \end{array}\right.$$
where λ is the intensity factor and $g_{3}(\nabla U)$ is the edge stop function based on small-target information enhancement.
(7)$$g_{3}(\nabla U)=1-\frac{1}{1-{\left(|\nabla U+(\int b/\mu )|/k\right)}^{2}}$$
where μ is the distance factor and *b* is the background contrast factor.
(8)$$\mu =||\vec{pq}||$$
where *p* is the target point and $p=(p_{x},p_{y})$; and *q* is the background point within a certain range of the target point and $q=(q_{x},q_{y})$.
(9)$$b=\left|\frac{U(p)+U(q)}{U(p)-U(q)}\right|$$
where U(p) is the grey scale value at point and *p* and U(q) is the grey scale value at point *q*.

#### 3.2.2. Target-Layer Information Enhancement

The classical mean filtering model is expressed as [24]:(10)$$\min_{U}\lambda |H|+\frac{1}{2}\|I-U\prime {\|}^{2}$$
where *I* is the enhanced image, U′ is the image to be enhanced after background suppression, λ is the regularization factor, and *H* is the mean curvature. The approximation operation is achieved by constructing the local directional curvature and calculating the center pixel to tangent plane distance to construct the filter, with the mean curvature regularized energy component calculated as follows [25]:(11)$$H\approx \frac{1}{8}\sum_{i=1}^{8}d_{i}=\left[\begin{array}{ccc} -\frac{1}{16} & \frac{5}{16} & -\frac{1}{16}\\ \frac{5}{16} & -1 & \frac{5}{16}\\ -\frac{1}{16} & \frac{5}{16} & -\frac{1}{16} \end{array}\right]⊗U\prime$$
where ⊗ is the convolution operation.

Calculation of the mean curvature projection operator:$$\begin{array}{l} 1.\;\mathrm{Input}\;\mathrm{image}\;U\prime (i,j)\\ 2.\;d_{1}=\frac{5}{16}(U\prime (i-1,j)+U\prime (i+1,j))+\frac{5}{8}U\prime (i,j+1)-\frac{1}{8}(U\prime (i-1,j+1)+U\prime (i+1,j+1))-U\prime (i,j)\\ 3.\;d_{2}=\frac{5}{16}(U\prime (i-1,j)+U\prime (i+1,j))+\frac{5}{8}U\prime (i,j-1)-\frac{1}{8}(U\prime (i-1,j-1)+U\prime (i+1,j-1))-U\prime (i,j)\\ 4.\;d_{3}=\frac{5}{16}(U\prime (i,j-1)+U\prime (i,j+1))+\frac{5}{8}U\prime (i-1,j)-\frac{1}{8}(U\prime (i-1,j-1)+U\prime (i-1,j+1))-U\prime (i,j)\\ 5.\;d_{4}=\frac{5}{16}(U\prime (i,j-1)+U\prime (i,j+1))+\frac{5}{8}U\prime (i+1,j)-\frac{1}{8}(U\prime (i+1,j-1)+U\prime (i+1,j+1))-U\prime (i,j)\\ 6.\;d_{\min}=\min\left\{\left|d_{1}\right|,\left|d_{2}\right|,\ldots ,\left|d_{4}\right|\right\}\\ 7.\;\mathrm{Output}\;\mathrm{image}\;I(i,j)=U\prime (i,j)+d_{\min} \end{array}$$

However, the median curvature filter performs poorly when dealing with information from complex backgrounds, mainly because only the mean curvature obtained from 8 triangular tangent planes in 8 neighborhoods is considered, and the classical triangular tangent planes are Figure 4:

The traditional mean curvature filtering algorithm uses a projection operator that fits a local surface with a minimum triangular tangent plane, which can achieve better image denoising and background suppression when dealing with simple background images, but the algorithm is not robust under complex background conditions, and although it can maintain the edge details of the image, there is a serious phenomenon of inadequate background suppression. Therefore, this paper considers the projection operator by extending the enhanced projection operator to calculate the minimum distance of each central pixel to the neighborhood tangent plane and updating the current central pixel using the minimum distance. The use of the implicit calculation of the curvature in curvature filtering assists the algorithm to converge quickly. It also achieves the objective of improving the algorithm’s performance in suppressing the background on top of the background.

The extended augmentation operator considers the mean curvature obtained from 12 trigonometric tangent planes in the 8-neighbourhood, with the trigonometric tangent planes shown in Figure 5.

The improved calculation of the mean curvature projection operator is as follows:$$\begin{array}{l} 1.\;\mathrm{Input}\;\mathrm{image}\;U\prime (i,j)\\ 2.\;d_{1}=\frac{5}{16}(U\prime (i-1,j)+U\prime (i+1,j))+\frac{5}{8}U\prime (i,j+1)-\frac{1}{8}(U\prime (i-1,j+1)+U\prime (i+1,j+1))-U\prime (i,j)\\ 3.\;d_{2}=\frac{5}{16}(U\prime (i-1,j)+U\prime (i+1,j))+\frac{5}{8}U\prime (i,j-1)-\frac{1}{8}(U\prime (i-1,j-1)+U\prime (i+1,j-1))-U\prime (i,j)\\ 4.\;d_{3}=\frac{5}{16}(U\prime (i,j-1)+U\prime (i,j+1))+\frac{5}{8}U\prime (i-1,j)-\frac{1}{8}(U\prime (i-1,j-1)+U\prime (i-1,j+1))-U\prime (i,j)\\ 5.\;d_{4}=\frac{5}{16}(U\prime (i,j-1)+U\prime (i,j+1))+\frac{5}{8}U\prime (i+1,j)-\frac{1}{8}(U\prime (i+1,j-1)+U\prime (i+1,j+1))-U\prime (i,j)\\ 6.\;d_{5}=\frac{5}{8}(U\prime (i,j-1)+U\prime (i+1,j))-\frac{1}{8}U\prime (i+1,j-1)-\frac{1}{16}(U\prime (i-1,j-1)+U\prime (i+1,j+1))-U\prime (i,j)\\ 7.\;d_{6}=\frac{5}{8}(U\prime (i,j+1)+U\prime (i-1,j))-\frac{1}{8}U\prime (i-1,j+1)-\frac{1}{16}(U\prime (i-1,j-1)+U\prime (i+1,j+1))-U\prime (i,j)\\ 8.\;d_{7}=\frac{5}{8}(U\prime (i,j-1)+U\prime (i-1,j))-\frac{1}{8}U\prime (i-1,j-1)-\frac{1}{16}(U\prime (i-1,j-1)+U\prime (i+1,j+1))-U\prime (i,j)\\ 9.\;d_{8}=\frac{5}{8}(U\prime (i,j+1)+U\prime (i+1,j))-\frac{1}{8}U\prime (i+1,j+1)-\frac{1}{16}(U\prime (i-1,j+1)+U\prime (i+1,j-1))-U\prime (i,j)\\ 10.\;d_{9}=\frac{5}{8}(U\prime (i-1,j)+U\prime (i,j-1))-\frac{1}{8}(U\prime (i-1,j+1)+U\prime (i-1,j-1))-U\prime (i,j)\\ 11.\;d_{10}=\frac{5}{8}(U\prime (i+1,j)+U\prime (i,j-1))-\frac{1}{8}(U\prime (i+1,j+1)+U\prime (i-1,j-1))-U\prime (i,j)\\ 12.\;d_{11}=\frac{5}{8}(U\prime (i+1,j)+U\prime (i,j+1))-\frac{1}{8}(U\prime (i+1,j+1)+U\prime (i+1,j-1))-U\prime (i,j)\\ 13.\;d_{12}=\frac{5}{8}(U\prime (i-1,j)+U\prime (i,j+1))-\frac{1}{8}(U\prime (i-1,j+1)+U\prime (i+1,j+1))-U\prime (i,j)\\ 14.\;d_{13}=\frac{5}{8}(U\prime (i-1,j)+U\prime (i,j-1))-\frac{1}{8}(U\prime (i-1,j-1)+U\prime (i+1,j-1))-U\prime (i,j)\\ 15.\;d_{14}=\frac{5}{8}(U\prime (i-1,j)+U\prime (i,j+1))-\frac{1}{8}(U\prime (i-1,j-1)+U\prime (i-1,j+1))-U\prime (i,j)\\ 16.\;d_{15}=\frac{5}{8}(U\prime (i+1,j)+U\prime (i,j+1))-\frac{1}{8}(U\prime (i-1,j+1)+U\prime (i+1,j+1))-U\prime (i,j)\\ 17.\;d_{16}=\frac{5}{8}(U\prime (i+1,j)+U\prime (i,j-1))-\frac{1}{8}(U\prime (i+1,j-1)+U\prime (i+1,j+1))-U\prime (i,j)\\ 18.\;d_{\min}=\min\left\{\left|d_{1}\right|,\left|d_{2}\right|,\ldots ,\left|d_{16}\right|\right\}\\ 19.\mathrm{Output}\;\mathrm{image}\;I_{A}(i,j)=U\prime (i,j)+d_{\min} \end{array}$$

### 3.3. B-Channel Air Domain, Frequency Domain Overall Enhancement

In this section, we retain information according to the different characteristics of the infrared small targets in a single infrared small-target image, with the final aim of rejecting the background.

#### 3.3.1. Band-Pass Filtered Frequency Domain Background Rejection

This sub-section will initially process the initial image based on the frequency-domain characteristics of the small target. As the infrared image has no color information, the brightness and texture detail are poor. In the case where the contrast between the real target and the background is not very strong, it is difficult to obtain good results by just transforming the phase spectrum to a plane, especially if the IR image is against a complex background. The exponential filter has a smoother filter band [15], and the image is smoothed without “ringing”.
(12)$$H(u,v)=e^{-{\left[\frac{D_{0}}{D(u,v)}\right]}^{n}}$$
where H(u,v) is the exponential filter transfer function; D_0_ is the cut-off frequency; D(u,v) is the distance from a pixel (u,v) to the centre of the frequency rectangle; and *n* is the order.

Given that the intensity distribution of a real infrared small target roughly satisfies a Gaussian distribution and that the Fourier transform of the Gaussian function results in itself, we can conclude that the infrared small target occupies a wide range in the frequency domain. The cut-off frequency should therefore not only ignore random noise and a smoothed background but also cover the entire frequency interval occupied by the target.

Therefore, it makes more sense to use bandpass filtering [26], so we combine two exponential filters with the following filter model:(13)$$H(u,v,D_{1},D_{2})\prime =H_{1}(u,v)-H_{2}(u,v)=e^{-{\left[\frac{D_{1}}{D(u,v)}\right]}^{n}}-e^{-{\left[\frac{D_{2}}{D(u,v)}\right]}^{n}}$$
where $H(u,v,D_{1},D_{2})\prime$ is the filter transfer function and *D*_1_ and *D*_2_ are different frequencies. Clearly, the combined bandwidth can completely cover the spectrum occupied by the small infrared target. The 3D diagram of the transfer function is as follows Figure 6:

Let the result of the frequency domain channel enhancement be $S_{U}$, and the expression be as follows:(14)$$S_{U}(u,v)=\left|U(u,v)⊗H(u,v)\prime \right|$$
where U(u,v) is the two-dimensional Fourier transform information of the original image and ⊗ is the convolution operator.

The reconstructed infrared small target image $s_{U}(u,v)$ is
(15)$$s_{U}(u,v)=\frac{1}{2\pi }\iint \left|U(u,v)⊗H(u,v)\prime \right|dudv$$

#### 3.3.2. Improved Double-Layer Local Contrast Null-Field Background Suppression

After the original image has been processed in the previous subsection, the complex background is further weakened, at which point the local contrast near the small target is further expanded. Inspired by the human eye’s visual system [27], we improved the MPCM (Multiscale patch-based contrast measure) [28] algorithm in this subsection by improving the local sliding window for small-target extraction in IR images. The nested structure is shown in Figure 7.
(16)$$D_{3}=\frac{1}{3^{2}}\sum_{j=1}^{3^{2}}I_{i}^{j}$$
(17)$$D_{9}=\frac{1}{9^{2}-3^{2}}\sum_{j=1}^{9^{2}-3^{2}}I_{i}^{j}$$
where *D*_3_ is the grey scale feature of the central region of the small target and *D*_9_ is the grey scale feature of the background region of the small target.

The local contrast *C* for small targets and complex environments is therefore expressed as
(18)$$C=\left|\frac{D_{3}-D_{9}}{D_{3}+D_{9}}\right|$$

The output of this subsection *I_B_* is therefore
(19)$$I_{B}=s_{F}⊗C$$

### 3.4. Dual-Channel Fusion Based on Weighted Superposition

The two-channel fusion method with weighted superposition has the advantage of being a simple and fast algorithm. Once the initial small-target enhancement was carried out for channels A and B, the following equation was used to weight the enhanced images of the two channels, and then the calculated result was used instead of the enhanced image of channel A to form the fusion result.
(20)$$\left\{\begin{array}{l} F=(1-w)I_{A}+wI_{B}\\ w=\frac{\sigma -\sigma_{\min}}{\sigma_{\max}-\sigma_{\min}}(c_{2}-c_{1})+c_{1} \end{array}\right.$$

*I_A_* and *I_B_* denote the enhanced images of channel A and channel B, respectively, and *F* is the result of their fusion; *w* is the weighting factor; σ is the mean value of the difference between the small target and the background after enhancement by channel B, $\sigma_{\min}$ and $\sigma_{\max}$ are the minimum and maximum values of the difference between each small target and the background; *c*_1_ and *c*_2_ are the minimum fusion weights of channel A and the maximum fusion weights of channel B *c*_1_ and *c*_2_ are the minimum fusion weights for channel A and the maximum fusion weights for channel B. Generally, *c*_1_ takes the value of 0.3~0.6 and *c*_2_ takes the value of 0.6~0.85.

## 4. Experiments and Results

Our algorithm was implemented on a common laptop computer, thus verifying that the IR motion small-target enhancement method in complex backgrounds has good robustness, high accuracy and a low false-alarm rate.

### 4.1. Sequential Experiments

In this section, we selected several images of weak infrared targets against complex backgrounds that were processed and analyzed using our method. The original images are shown below. Figure 8a shows a field ground background with a high concentration of reeds and a particularly complex edge shape, and a distant lake shore with small islands and unidentified protrusions, which are particularly prone to false alarms, and at the same time, small targets are interfered with by high-intensity and high-complexity flooding and are not easily detected. Figure 8b shows the classic aerial background with a more concentrated cloud distribution and stronger intensity, making it easy to generate false alarms.

For the above test images, the intermediate steps of our proposed method are shown in Figure 9 and Figure 10. Each image consists of 10 images: (a) the original image to be processed; (b) the pre-processed image with closed operations; (c) the image after the improved PM model processing in channel A, where the background information in the image is drastically weakened; (d) the image where the background is further weakened by the improved curvature filtering processing in channel A; (e) the output small target position in channel A; and (f) the band-pass filtering in channel B. The processing result (g) is the B channel double-layer local contrast processing result, (h) is the B channel processing result, (i) is the A-and-B-blended target-position extraction image, and (j) is the enhancement result, where the green box is the false alarm that appeared during the enhancement process and was finally eliminated, and the red box is the enhanced small infrared target.

Figure 11 shows some typical infrared images in the dataset and their processed images. Six of the images are a bright cloud background, a bright-haze weather background, a complex man-made building background, a field background and a complex building background on a riverbank. The targets in the field background and the complex building background on the riverbank are dark targets, and the others are bright targets.

### 4.2. Comparative Experiments with Different Methods

#### 4.2.1. Evaluation Criteria

In order to evaluate the advantages and disadvantages of image-processing algorithms intuitively, we used SCR (Signal-to-Clutter Ratio) [29], SCRG (Signal-to-Clutter Ratio Gain) [29], BSF (Background Suppress Factor) [30], Pd (Probability of detection) [30], Fa (false rate) [30] and ROC (receiver operating characteristic curve) [6] to describe the quality of the enhancement method.

The SCR is a basic indicator to measure the significance of the target and is also a measure of the difficulty of image detection. The larger the indicator, the more obvious the processing effect, which is defined as Formula (21):(21)$$SCR=\frac{\left|\mu_{t}-\mu_{b}\right|}{\sigma_{b}}$$
where $\mu_{t}$ represents the average pixel value of the target, and $\mu_{b}$ and $\sigma_{b}$ represent the average pixel value and pixel standard deviation of the background surrounding the target, respectively.

SCRG reflects the degree to which the input and output of the target are enhanced relative to the background and can also be used to describe the difficulty of small-target detection.
(22)$$SCRG=\frac{SCR_{out}}{SCR_{in}}$$
where *SCR_out_* and *SCR_in_* stand for the SCR values of the output and input images.

BSF reflects the background suppression ability of the detection method. A higher BSF means that the detection algorithm can suppress background clutter more effectively, which is defined as Formula (23):(23)$$BSF=\frac{\sigma_{in}}{\sigma_{out}}$$
where $\sigma_{in}$ and $\sigma_{out}$ represent the local background standard deviation in the original image and the local background standard deviation in the processed image, respectively.

*Pd* measures the accuracy of target detection by comparing the detection result with the true value, which is defined as Formula (24):(24)$$Pd=\frac{Np}{Nr}$$
where the number of detected targets is *Np* and the number of true values is *Nr*.

*Fa* obtains the result by calculating the proportion of the pixel to all the pixels in the image, which is defined as Formula (25):(25)$$Fa=\frac{Pf}{Pa}$$
where *Pf* is the error prediction pixel and *Pa* is all the pixels in the image.

In the ROC curve of dim-target detection, *Pd* is the horizontal axis and *Fa* is the vertical axis. Generally, when the *Fa* is the same, the higher the *Pd*, the better the performance of the algorithm.

#### 4.2.2. Experimental Method

In this simulation, we used a set of real infrared images with the original noise and target removed as the original sequence shown in Figure 12.

Firstly, we added an artificial target of size 44 with an intensity satisfying a Gaussian distribution to each image. Next, Gaussian noise with a mean of 0 and different variances was added for enhancement, as shown in Figure 13.

The following in Table 1 shows the average evaluation indicators of our method, Top-hat [31], TDLMS [32] and High-pass [33] when the SNR is 2.

It can be clearly seen from the table that among all the methods, our method had a higher Pd and lower Fa, which was evidently better than other methods.

Figure 14 shows the ROC curves corresponded to the enhancement rate and false alarm rate for our method, Top-hat, TDLMS and High-pass, at a signal-to-noise ratio of 2.

The graph shows that our method reaches 100% detection probability significantly faster due to other methods and has lower false alarms. Experiments have shown that the method has a good ability to eliminate false alarms while maintaining a stable enhancement rate in low-signal-to-noise-ratio infrared images.

### 4.3. Discussion

For dim-target detection in a complex background, we established a two-channel detection method. In the process of establishing dual-channel detection, we made use of the characteristics of small targets in complex infrared images, establish the imaging principle and the mapping relationship of different stages, and simplify the processing process. According to the formula, the error superposition caused by double-channel detection is effectively avoided. The method we used was faster and more effective than traditional methods.

When we improved the PM filter model, we used a new diffusion function, which made it possible for the PM model to process small-target images. Even for highly complex images, it took only a few iterations to get the target. It was found that the improved curvature filter could be used to enhance the image effectively.

To verify the effectiveness of the algorithm, we adopted the method of adding target and noise. With the decrease in image SNR, when the SCRG of top-hat method was reduced to 1.056, our index remained at 3.143.

During the experiment, we found that there were still some missing targets that were not detected, so the digital modeling of complex backgrounds and small targets in complex backgrounds needs to be further strengthened in the future. After obtaining better model information, we can better analyze the reasons for missing targets and improve the algorithm.

At present, our methods were all based on CPU. In the future work, we expect to use FPGA (Field Programmable Gate Array) to process information and realize image parallel processing and algorithm optimization through hardware platforms to improve the timeliness of image processing. In addition, the algorithm had a low computational complexity and a single detection sensor was cheap, so it can be deployed in a large range in the field to effectively detect unknown illegal aircraft.

## 5. Conclusions

Infrared motion small-target enhancement against complex backgrounds has been a difficult task previously due to the lack of texture and shape information, cloud-edge interference, and low target-to-background contrast. In this paper, a two-channel operation enhancement method was proposed. A closed operation was used to pre-process the image. Then, a two-channel enhancement was applied, with an improved curvature filter in channel A first to weaken the background, while an improved PM model was used to enhance the target for overall enhancement, and a modified self-bandpass filter in channel B for coarse enhancement, followed by a two-layer local-contrast algorithm for fine enhancement, based on which, two-channel fusion based on weighted superposition was performed.

In particular, the authors provided the sensitivity parameter, that is, the signal-to-noise ratio for image processing with a complex background. Many experiments based on real and simulated infrared images have shown that the SCRG, BSF and Pd can reach 3.143, 2.161 and 0.99 under different complex scenes, and the proposed method has good robustness under different SNRs. Compared with other methods, our method significantly outperformed them in terms of signal-to-noise-ratio enhancement for processing images with complex backgrounds. The problem of improving the signal/noise ratio of small-sized infrared moving targets on a complex background has been solved. The effectiveness of this method for the comprehensive improvement of the image of an infrared moving target was confirmed.

## Figures and Tables

**Figure 1 sensors-23-06215-f001:**
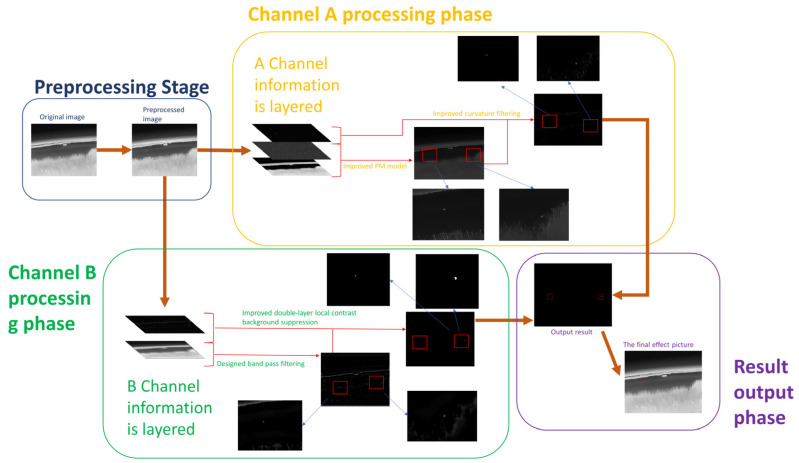
Enhanced flow chart.

**Figure 2 sensors-23-06215-f002:**
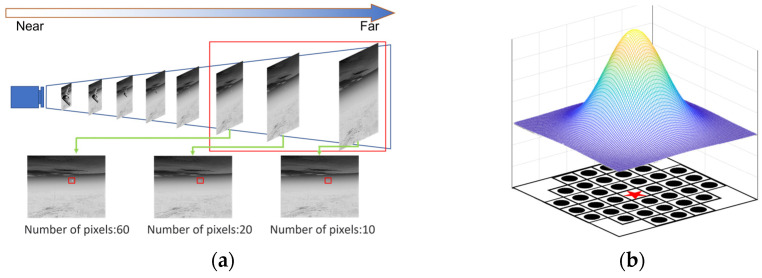
Small-target structure and closed operator structure elements. (**a**) Small-target imaging principle. (**b**) Structural elements constructed according to general small-target projection.

**Figure 3 sensors-23-06215-f003:**
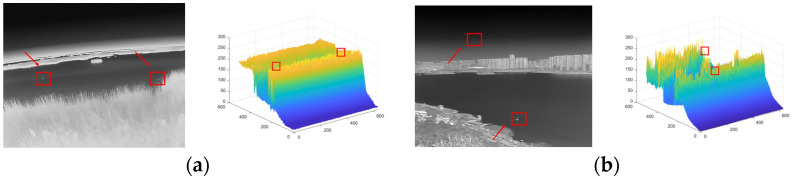
A small target in a complex context. (**a**) Aims in a complex context a. (**b**) Aims in a complex context b.

**Figure 4 sensors-23-06215-f004:**
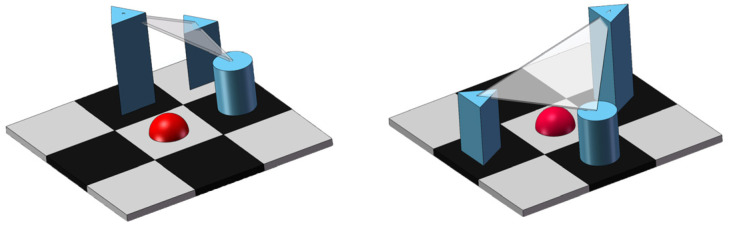
Classic triangular cut.

**Figure 5 sensors-23-06215-f005:**
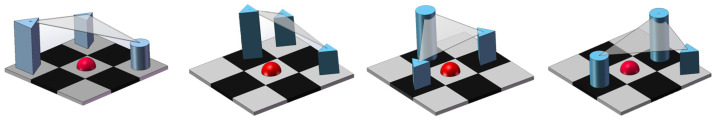
Improved triangular cut.

**Figure 6 sensors-23-06215-f006:**
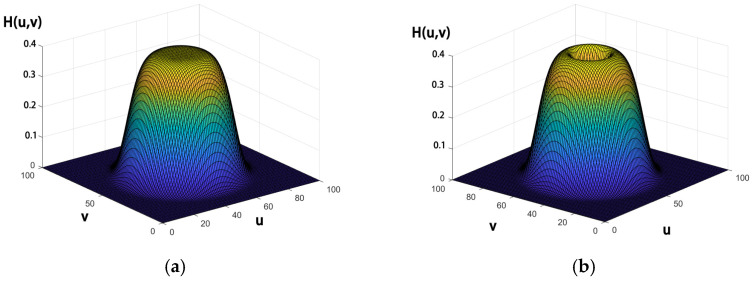
3D transfer functions for the exponential and band-pass filters. (**a**) 3D transfer function for exponential filters. (**b**) 3D transfer function for bandpass filters.

**Figure 7 sensors-23-06215-f007:**
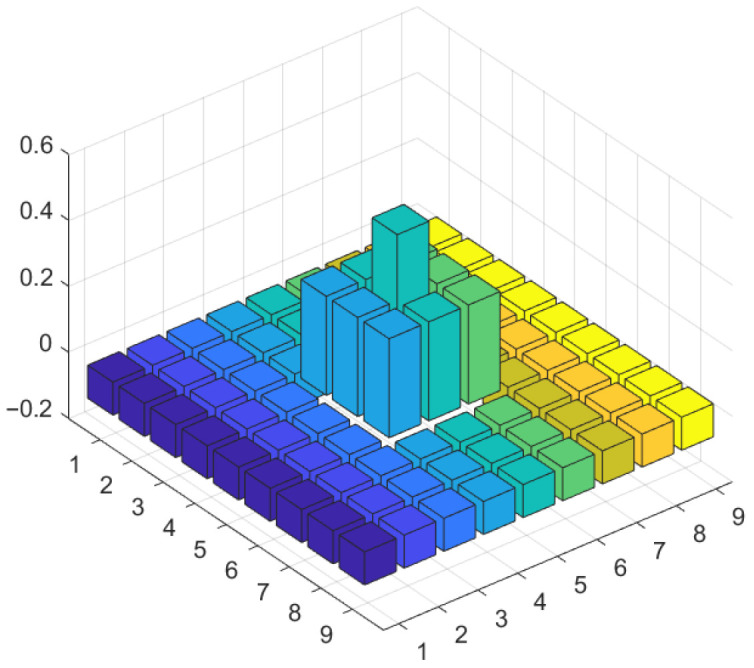
Nested structures.

**Figure 8 sensors-23-06215-f008:**
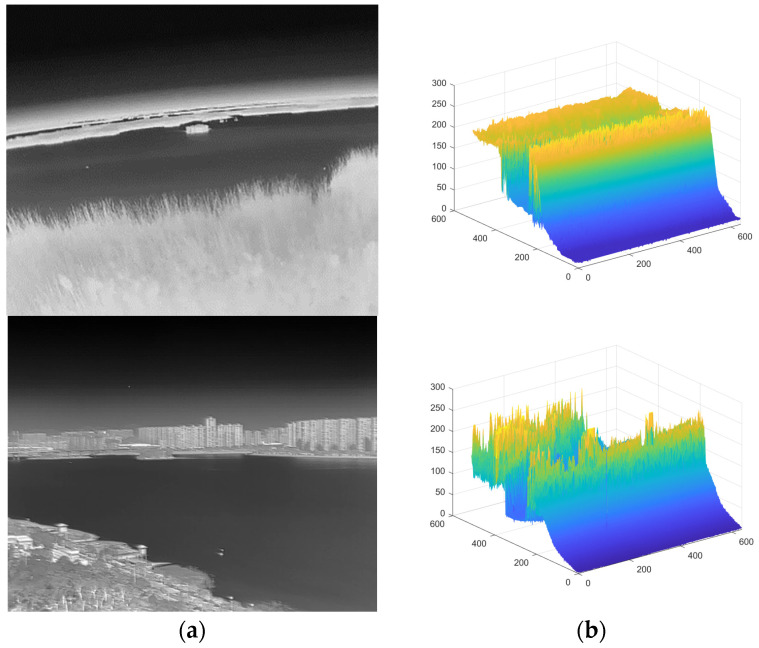
Image of a small infrared target against a complex background to be processed. (**a**) Field ground background and ground target salient map. (**b**) Aerial background and aerial target salient prominence.

**Figure 9 sensors-23-06215-f009:**
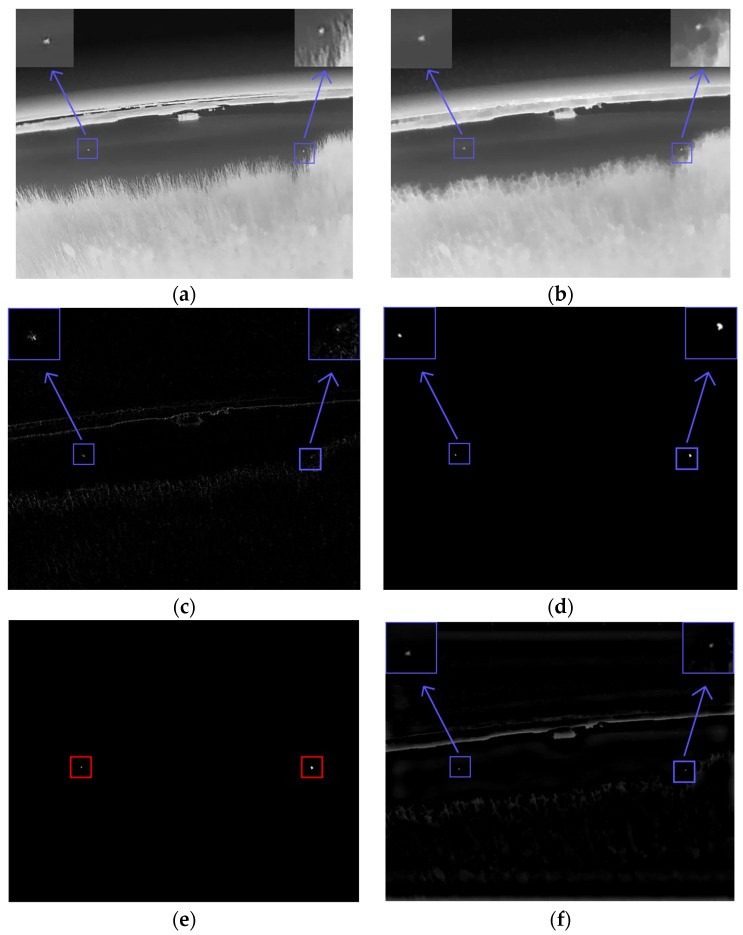
Wetland background infrared small-target processing. (**a**) Original image to be processed. (**b**) Pre-processed images. (**c**) Improved image after PM model processing. (**d**) Image processed using improved curvature filtering. (**e**) Channel A outputs small target positions. (**f**) Bandpass filtering results. (**g**) Double-layer local contrast processing results. (**h**) Channel B processing results. (**i**) Enhanced results. (**j**) Significant graphs of enhanced results.

**Figure 10 sensors-23-06215-f010:**
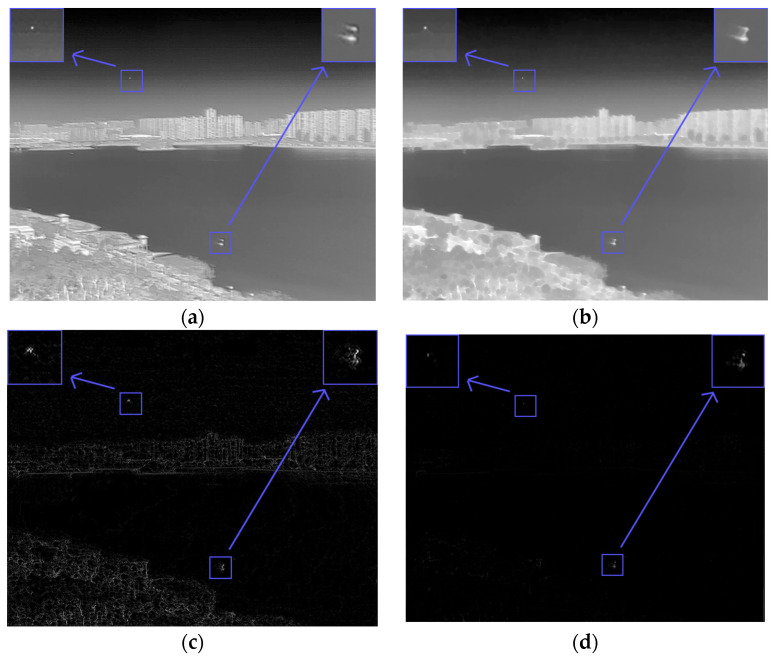
Processing of small shore-side background infrared targets. (**a**) Original image to be processed. (**b**) Pre-processed images. (**c**) Improved image after PM model processing. (**d**) Image processed using improved curvature filtering. (**e**) Channel A outputs small-target positions. (**f**) Bandpass filtering results. (**g**) Double-layer local contrast processing results. (**h**) Channel B processing results. (**i**) Enhanced results. (**j**) Significant graphs of enhanced results.

**Figure 11 sensors-23-06215-f011:**
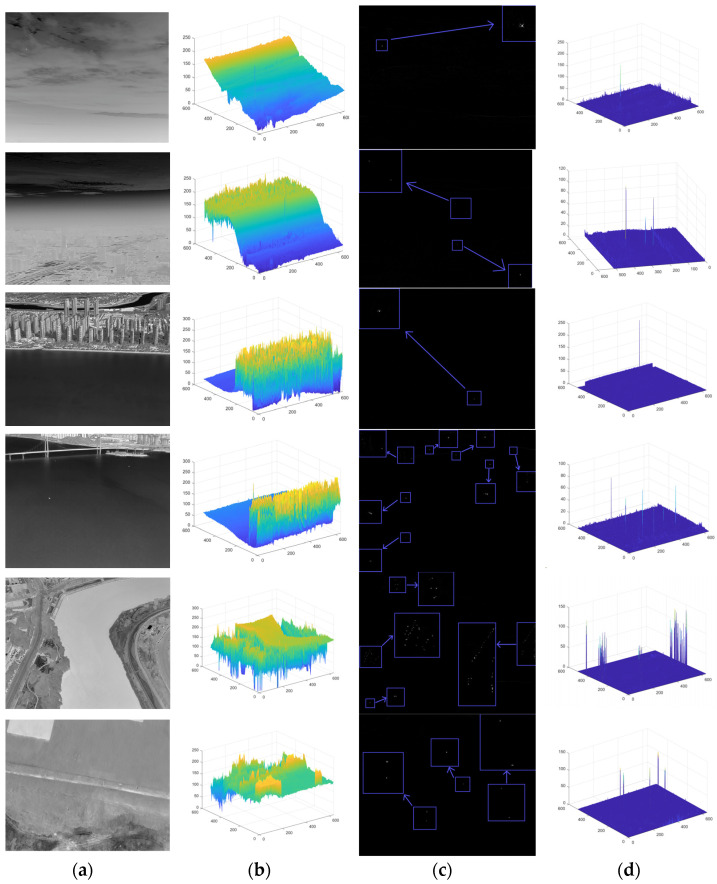
A typical infrared image and its processed image. (**a**) Original image. (**b**) Original image target prominence map. (**c**) Processing results graph. (**d**) Targeted significance plot of treatment results.

**Figure 12 sensors-23-06215-f012:**
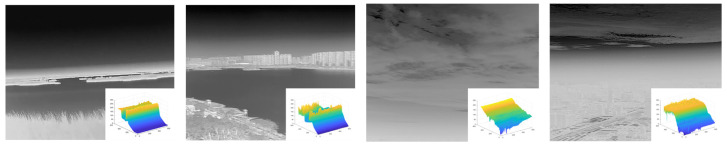
Noise-free target-free map.

**Figure 13 sensors-23-06215-f013:**
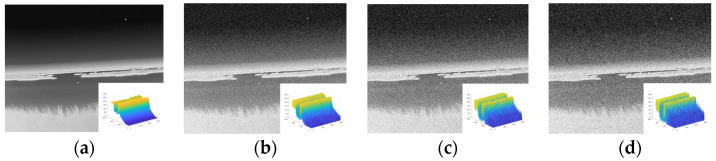
Adding targets and noise maps. (**a**) Add target only. (**b**) Noise variance of 0.01. (**c**) Noise variance of 0.02. (**d**) Noise variance of 0.03.

**Figure 14 sensors-23-06215-f014:**
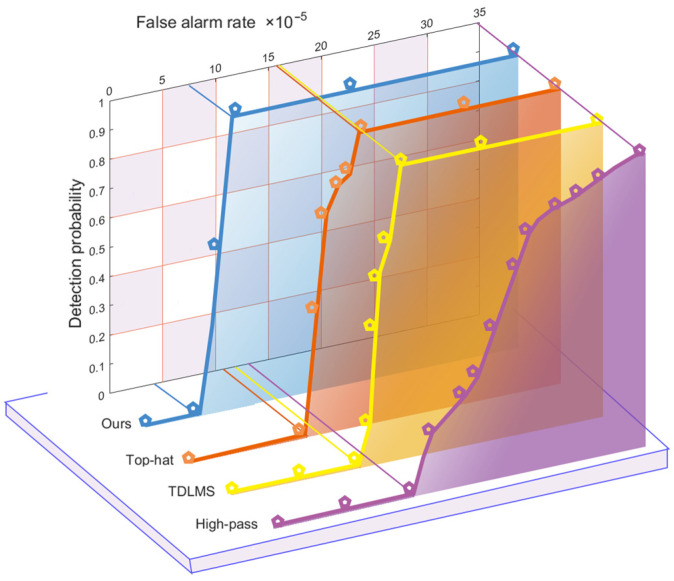
ROC curves for different methods.

**Table 1 sensors-23-06215-t001:** Average metrics of the 4 methods.

	**Top-Hat**	**TDLMS**	**High-Pass**	**Our**
*SCRG*	1.056	2.242	2.419	3.143
*BSF*	1.322	1.407	1.555	2.161
*Pd*	0.730	0.821	0.865	0.993
* Fa *	0.315	0.645	0.565	0.079

## Data Availability

Data is not available due to privacy restrictions.

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
