# Peer review of "Infrared Image-Enhancement Algorithm for Weak Targets in Complex Backgrounds"

_sensors, 2023, doi:10.3390/s23136215_

Round 1
Reviewer 1 Report
This paper mainly proposed a new enhancement method for small infrared moving targets in complex environments, which used Digital Image Processing methods, filtering methods, image fusion, etc. However, there is still much room for improvement and the author is advised to make improvements before publishing.
1. Paper "Infrared dim target detection method inspired by human vision system" proposed a Spatial significant area extraction method to depress background and enhance infrared dim target; Paper "Infrared Sensation-Based Salient Targets Enhancement Methods in Low-Visibility Scenes" proposed different enhancement method when dealing with inclement weather and their detection performances. How does the enhancement method used in this article differ from the enhancement methods used in these articles and what is your innovation?
2. It is suggested that the article be enriched by the detection of small and dim infrared targets in complex environments.
3. Lack of actual data comparison and elaboration of experimental results using common image quality evaluation indicators, for example, the Peak Signal-to-Noise Ratio (PSNR), Structure Similarity Index Measure (SSIM), Mean Squared Error (MSE), ROC curves, etc. In this paper, how can you tell when an infrared target has been effectively enhanced?
4. Figure 1: The arrows are not clear, e.g. in the enhancement phase of band-pass filtering and PM model, what the red arrow is doing? I hope the author will explain and improve the framework diagram.
5. Formula: The explanation of the parameters of several formulae is missing punctuation.
6. Eq(6):
What the “I” refers to here is not explained in the text.
7. Section 3.2.2: Why was the Mean Filtering chosen directly? Why not choose Gaussian filtering or other filtering?
8. Figures: Almost all Figures are not cited in the text, e.g. Figure 6,...,Figure 10, etc.
9. There are many unexplained capitalized proprietary names in the text, e.g. PM, ROC, etc.
10. Eq(21)(22): What are “Pf” and “Pd” and the formulas are not quoted in the text and the calculation results are not visible, please explain.
11. Section 4.3: “our method decreases slowly as the signal-to-noise ratio of the image decreases, but the method still works effectively in low signal-to-noise environments”, does the low signal-to-noise environments refer to complex environments? I don't see any more test results for different environments in the article.
12. English expression throughout the text needs to be improved and punctuation needs to be checked.
English expression throughout the text needs to be improved and punctuation needs to be checked. The author is advised to make extensive revisions before publishing.
Author Response
Thank you for your suggestions. We have revised the article. Thanks again.
Please see the attachment.

Reviewer 2 Report
1. A study by Yingchao Li et al. "An infrared image enhancement algorithm for weak targets on a complex background" is devoted to mathematical image processing. The article is devoted to the detection of small thermal targets. In particular, this applies to the infrared range. Identifying targets in the infrared range is a difficult task, especially in conditions of a complex background. This problem is relevant in the fields of technology - these are infrared early warning systems, precision targeting systems, and satellite remote sensors. The problem is this: when moving, the contrast between the subject and the background changes significantly depending on the type of subject and the distance of the image. The presence of these factors in an IR image can cause significant obstacles to the detection of small targets.
The problem of the signal/noise ratio of small-sized infrared moving targets on a complex background has been solved. The effectiveness of this method for comprehensive improvement of the image of an infrared moving target has been confirmed.
2. There are traditional methods and algorithms for image processing of infrared small targets. They are based mainly on maximum median filtering algorithms, wavelet transform algorithms, etc. These algorithms have a good effect, but have a high error rate for infrared images with complex backgrounds. A problem arises if the size of the infrared target to be detected is too small.
3. The article proposes a new method of complex signal amplification from small-sized infrared moving targets. Detection consists in extracting the background based on two-channel information. Traditional uniform filtering tends to remove noise while blurring the edges of the image. At the same time, there is a loss of image detail. To improve image quality, a nonlinear diffusion method was developed. A three-dimensional diagram of the transfer function is constructed. In particular, the authors emphasized the use of exponential and bandpass filters.
*The work is written clearly and distinctly.
*Graphs and formulas are clear and informative.
Notes for correction:
* It is not clear - why are there no references in the text to the literature given at the end of the article (example, [1, 2] and so on)?.
* p.4 eq(6)-----------g3 -g3?
*The font of some characters in the text is much smaller than the main letters. It should be fixed.
5. The following can be attributed to the conclusions. A two-channel method of image quality improvement is proposed. Image preprocessing was performed and two-channel image enhancement was applied. In particular, it is: alternately with a background reduction filter and a two-layer local contrast algorithm. In particular, the authors provide the sensitivity parameter, that is, the signal-to-noise ratio for image processing with a complex background. The experiments conducted by the authors showed that the method has high efficiency, speed, quality and good reliability. The problem of improving the signal/noise ratio of small-sized infrared moving targets on a complex background is solved. The effectiveness of this method for comprehensive improvement of the image of an infrared moving target has been confirmed.
6. The necessary literature is indicated - 30 items. Links are new.
7. Figures 3, 8, 9, 10, 11, 12, 13 seem to be small, but in this combination they are informative and quite readable. There are no tables. True, there are inserted equations in the form of a matrix - they are quite informative
Conclusion:
The work can be published after minor corrections.
Author Response

(The authors gave the same response as above.)

Reviewer 3 Report
The authors propose algorithms for detecting small targets in the infrared range. The proposed algorithm makes it possible to obtain payoffs in comparison with the known ones. The article may be useful to specialists in image and signal processing. However, there are a number of remarks about the work:
1) In the review part of the work, related results are mentioned, but no references are given. I would also highly recommend adding articles related to image modeling for signal filtering and detection (doi.org/10.1134/S105466181901005X, https://ceur-ws.org/Vol-2711/paper39.pdf, doi: 10.18287/2412- 6179-CO-922)
2) In the abstract, it is necessary to disclose in detail the connection of research with the topics of the Sensors journal.
3) It is not clear where the matrix coefficients in formula (11) come from.
4) There is a lot of empty space on page 6.
5) It is necessary to refer to figures with numbering everywhere, and also to bring the article to the style in accordance with the design requirements
6) Figure 11 should be placed on one page
7) In Figure 14, it is not clear what is plotted along the X axis. The numerical values of the area under the ROC curve are not shown.
Author Response

(The authors gave the same response as above.)

Reviewer 4 Report
Comments of this reviewer on the manuscript Sensors-2436868 are as follows:
1. This manuscript proposes an enhanced method for small infrared moving targets based on two-channel information. Novelties are evident, but the manuscript is not well organized. There are some comments that the Authors should address.
2. The idea is not original, and scientific contributions must be highlighted.
3. There is a list of references at the end of the manuscript, but the Authors did not cite any reference in the manuscript. Why? This style of writing scientific articles is uncommon. For example, in the following sentence: “Traditional infrared small target enhancement algorithms mainly include Top-Hat, Max-Mean, and Max-Median filtering algorithms, wavelet transform algorithms, etc.”, each of the methods should be followed by an appropriate reference. Likewise, when mentioning the names of authors such as “Zhang et al.” and “LIN et al.”, the appropriate reference should be provided.
4. In the manuscript, the abbreviation “ROC” was used, for which no explanation was given. Not all readers know what “ROC” means. In addition, there are some other abbreviations in the manuscript that are not defined.
5. In the introduction, the literature review is very poor and must be improved in terms of the relevant state-of-the-art. The research gap must be identified within the introduction as well.
6. Keywords should be listed in alphabetical order.
7. At the beginning of Section 2, the Authors stated the following: “Our approach consists of five main steps…” However, below Figure 1, readers can find descriptions of only four main steps.
8. The quality of the English language should be improved. There are errors and ambiguities in the manuscript such as “where the classical edge stop function, denoted as”, “we improve the MPCM algorithm” (What is MPCM? What is the source of the algorithm?), “The graph shows that our method reaches 100% enhancement rate significantly faster due to other methods and has lower false alarms.” (unclear sentence), “Electron.ics”, “tar-gets”, etc.
9. The descriptions/captures below the figures are sketchy and could be improved to better describe the content of the illustrations.
10. The discussion is short, does not refer to the specific results obtained and is almost general.
11. The conclusion has the form of an abstract, its content does not refer to specific results obtained (it only announces them in general) and does not contribute to the body of current knowledge in the field of research.
12. Reference list was not prepared in line with the guidelines for authors of the journal.
13. In the reference list, it seems that the references marked with the numerals “20.” and “21.” are the same.
14. In general, the presentation style must be changed and improved.
Moderate editing of English language required.
Author Response

(The authors gave the same response as above.)

Round 2
Reviewer 4 Report
The authors adequately answered all my comments and questions. However, there is one shortcoming that can be corrected during the proofreading phase. When citing references that have more than one author, they cannot be cited as follows: Zhang [14], Bae [15], Xie [16], Cai [17], LIN [18], etc. These citations should be as follows: Zhang et al. [14], Bae et al. [15], Xie et al. [16], Cai et al. [17], Lin et al. [18], etc.